# Morality is Contextual: Learning Interpretable Moral Contexts from Human Data with Probabilistic Clustering and Large Language Models

## Abstract

Moral actions are judged not only by their outcomes but by the context in which they occur. We present COMETH (Contextual Organization of Moral Evaluation from Textual Human inputs), a framework that integrates a probabilistic context learner with LLM-based semantic abstraction and human moral evaluations to model how context shapes the acceptability of ambiguous actions. We curate an empirically grounded dataset of 300 scenarios across six core actions (violating *Do not kill*, *Do not deceive*, and *Do not break the law*) and collect ternary judgments (Blame/Neutral/Support) from $N=101$ participants. A preprocessing pipeline standardizes actions via an LLM filter and MiniLM embeddings with K-means, producing robust, reproducible core-action clusters. COMETH then learns action-specific *moral contexts* by clustering scenarios online from human judgment distributions using principled divergence criteria. To generalize and explain predictions, a Generalization module extracts concise, non-evaluative binary contextual features and learns feature weights in a transparent likelihood-based model. Empirically, COMETH roughly doubles alignment with majority human judgments relative to end-to-end LLM prompting ($\approx 60\%$ vs. $\approx 30\%$ on average), while revealing which contextual features drive its predictions. The contributions are: (i) an empirically grounded moral-context dataset, (ii) a reproducible pipeline combining human judgments with model-based context learning and LLM semantics, and (iii) an interpretable alternative to end-to-end LLMs for context-sensitive moral prediction and explanation.

## 1 Introduction

Artificial Intelligence (AI) is increasingly shaping important aspects of human activity, mediating not only practical tasks but also social, economic, and moral interactions. This expanding role amplifies the urgency of ethical supervision and moral alignment, in order to ensure that AI systems consistently comply with human ethical standards across diverse contexts (Awad et al., 2022; Rahwan et al., 2019; Gabriel, 2020; Khamassi et al., 2024). The growth in AI capabilities and applications, notably through Large Language Models (LLMs) and Reinforcement Learning (RL), increases the need for systems capable of adapting to morally complex situations that can be very sensitive to contextual variability (Bonnefon et al., 2024; Perez et al., 2023; Ma et al., 2024).

To meet this challenge, computational ethics has emerged as an interdisciplinary field that draws from moral philosophy, cognitive science, social psychology, law, and ethics to inform AI design with empirically informed ethical and computational frameworks grounded in our understanding of human moral cognition and behavior (Awad et al., 2022; Scherrer et al., 2023; Pflanzer et al., 2023). Central to this pursuit is recognizing the profound context-dependency of human moral cognition: identical actions frequently elicit divergent moral judgments depending on cultural norms, situational dynamics, or psychological nuances (Awad et al., 2020; 2018; Forbes et al., 2021).

Empirical studies, such as The Moral Machine experiment (Awad et al., 2018), reveal significant global variations in moral judgments, highlighting both universal ethical intuitions and culturally specific differences (Awad et al., 2020). These findings show the limitations of rigid ethical frameworks in capturing the flexible, context-sensitive nature of human morality, pushing for adaptive AI alignment methods that reflect this reality (Awad et al., 2024; Birhane et al., 2024; He et al., 2025).

Ensuring that AI-based systems, notably LLMs, remain beneficial to the society is a challenge that research on AI alignment intends to meet. AI alignment research focuses on the value alignment problem, which Russell & Norvig (2021) states as follows: "the values or objectives put into the machine must be aligned with those of the human". It is now consensual that an AI-based system should align with human values (Gabriel, 2020). A growing line of recent research investigates LLMs' ability to align with human values and make consistent moral decisions (Scherrer et al.,

2023; Garcia et al., 2024). Aligning LLMs with human values is often approached through reinforcement learning from human feedback (Ji et al., 2023), but faces the challenge of diverse and sometimes conflicting human inputs (Conitzer et al., 2024). However, it is difficult to ensure alignment with human values in opaque AI systems where no real reasoning abilities, nor guarantees of truth value can be ensured (Khamassi et al., 2024). A promising avenue to overcome the limitations of current approaches is to combine LLMs with systems such as model-based RL, capable of autonomously learning the effects of specific actions in the real world to enable (1) AI reasoning about actions' potential risk to undermine human values, and (2) AI explainability of why a particular action presents a specific risk.

Indeed, Model-Based Reinforcement Learning (MBRL) is particularly suited for addressing the challenge of context adaptation due to its intrinsic capacity to model structured, context-specific reward dynamics and generalize learned behaviors across varying scenarios (Rodriguez-Soto et al., 2022; Benechehab et al., 2025; Chartouny et al., 2025). Recent innovations leverage LLM-generated datasets of moral dilemmas to train MBRL agents, thus utilizing the contextual understanding and representational power of language models to enhance moral reasoning capabilities (Bano et al., 2023; Du et al., 2023; Ziems et al., 2022). However, such synthetic datasets often lack empirical grounding in real human moral behavior, raising critical concerns about their ecological validity, reproducibility, and the ethical implications of circumventing participant consent and excluding culturally grounded moral diversity.

In this paper, we present COMETH (Contextual Organization of Moral Evaluation from Textual Human inputs) a novel framework that integrates empirical moral judgment data with a Probabilistic RL architecture designed to infer context-specific reward models from ternary human moral evaluations (blame, neutral, support). Our methodology is evaluated on a dataset of 300 high-ambiguity moral scenarios that vary along multiple contextual dimensions. A key innovation of COMETH lies in its hybrid architecture, which combines the multi-model context-learning capacity of Model-Based Reinforcement Learning with the semantic abstraction of Large Language Models. We employ LLMs to extract features of states and actions from natural language moral scenarios. The context-learning RL agent then autonomously detects the need for context differentiation when the same (state, action) pair yields divergent distributions of human moral judgments. This mechanism enables the clustering of moral contexts and supports generalization across semantically related scenarios. This integration offers a more efficient, transparent and interpretable alternative to end-to-end LLM pipelines by explicitly modeling state, action, and reward representations. Through this framework, our aim is to advance the development of morally aligned systems capable of better responding to the nuanced and context-dependent nature of human moral reasoning.

## 2 THE COMETH PIPELINE

In this section, we present COMETH (Contextual Organization of Moral Evaluation from Textual Human inputs), a novel method that integrates Probabilistic Context Learning inspired from Model-Based Reinforcement Learning (MBRL) with Large Language Models (LLMs) and human moral evaluations to model how context shapes the acceptability of morally ambiguous actions (Figure 1). Our approach makes key contributions across four dimensions. **(i)** We introduce a Probabilistic Context Learner to cluster scenarios, enabling the agent to represent and distinguish moral contexts based on human-derived evaluative outcomes. **(ii)** We collect an empirically grounded dataset of **300** scenarios across six core actions with ternary judgments (blame/neutral/support) from **101** participants, grounded in Gert's common-morality rules (Gert, 2004). **(iii)** We develop a custom LLM-based pre-processing pipeline that abstracts scenario descriptions into structured embeddings of moral actions, enabling the agent to identify semantically similar actions across diverse narrative contexts. **(iv)** We add an interpretable Generalization module which extracts the key contextual features of the clusters and learns feature weights, markedly improving predictive alignment with human judgments vs. end-to-end LLM prompting.

### 2.1 PROBABILISTIC CONTEXT LEARNER

The Probabilistic Context Learner's objective is to autonomously infer and refine clusters online—referred to as moral contexts—by identifying patterns in ternary outcome distributions that reflect human moral judgments. These context models aim at capturing the normative variability observed in human societies, where actions are judged differently depending on the context (*e.g.,* punching as an aggressor vs. in a situation of self-defense).

Each scenario is represented as a triplet scenario, action, judgment series, where the judgment series encodes the moral evaluations (Blame, Neutral, Support) collected from the online survey (Section 2.2), and the core action is extracted via pre-processing (Section 2.3). The Probabilistic Context Learner groups scenarios for a given action into context models based on the distribution of these judgment series. For each action, the number of context models is smaller than the number of scenarios, as the agent clusters scenarios with similar moral judgments. Contexts are created

Figure 1: **COMETH Pipeline**. 101 participants answered an online survey which permit us to obtain moral judgments distributions for the 300 scenarios we generated. Then a pre-processing algorithm extracts the core action of the scenarios and group the scenarios sharing the same core action. The *Probabilistic Context Learner* then clustered scenarios with the same action into distinct moral contexts based on human judgment distributions, using adding and merging modules. To interpret and generalize, an LLM-based module extracted descriptive contextual features, which were binarized into feature vectors. Aggregate feature profiles were computed for each context, and feature weights were learned via a likelihood-based model. Finally, predictions of moral judgments were evaluated using a softmax-based scoring function. Colors in the figure represent different core actions.

and updated online as new scenarios are observed, using two main components: the *adding module* and the *merging module*, inspired from Chartouny et al. (2025) (See Section A.3).

When a new scenario is presented, the *adding module* determines whether it fits an existing context for the corresponding action or requires creating a new one. For each stored context, the agent compares the scenario's ternary moral judgment distribution to the context's reward distribution using the Kullback-Leibler (KL) divergence. A small constant $\epsilon$ ensures all distributions are well-defined, and values are normalized.

If the minimal KL divergence is below a threshold $\Delta_a$, the scenario is assigned to the closest context; otherwise, a new context is created. Assigning a scenario updates the context by adding its reward distribution to refine the model and appending the scenario to the set associated with the context.

As scenarios accumulate, some context models may become similar. To prevent redundancy, a *merging module* compares models using a semi-weighted Jensen-Shannon divergence (swJS), which accounts for both distribution similarity and relative context sizes. After adding a scenario or creating a new context, the agent computes swJS between all pairs of models for the action. If the divergence between two models falls below a threshold $\Delta_m$, the models are merged, consolidating their observations and reducing the total number of contexts while preserving a diverse representation of moral scenarios.

We determined that the optimal threshold values are $\Delta_a = 0.12$ and $\Delta_m = 0.03$. Details regarding the parameter search procedure can be found in the appendix (see Section A.4).

## 2.2 Empirical Collection of Human Moral Judgments

The objective of COMETH is to investigate how variations in contextual information modulate the moral evaluation of a given action. Specifically, we hypothesize that a single action may be perceived as morally permissible in one context while being judged morally impermissible in another. To empirically assess this hypothesis, we collected normative data on human moral judgments through an online survey.

We constructed a dataset of 300 scenarios, inspired by Scherrer et al. (2023) and grounded in Gert's common morality framework (Gert, 2004). While Gert's framework encompasses ten rules, we focused on three—Do not kill, Do not deceive, and Do not break the law—and derived six core actions: euthanasia, killing in protection, lying for support, lying for self-interest, stealing, and engaging in illegal protest. Each action was expanded into 50 impersonal scenario variants using prompting with GPT-4 and manual rewriting, ensuring that participants evaluated the morality of others' actions rather than their own decisions.

The final survey ($N = 101$, mean age = 35.2, 48 women) asked participants to judge each action as Blame, Support, or Neutral. To control for order and language effects, scenarios were randomized across six groups and presented in English or French. Further methodological details, scenario variants, and demographic breakdowns are provided in the Appendix (see Section A.1).

## 2.3 Pre-processing Methods

The Probabilistic Context Learner requires consistent representations of moral actions to avoid conflating distinct actions that elicit similar judgments or splitting semantically equivalent ones. To achieve this, scenarios were pre-processed with LLM-based filtering (Mistral-7B, Llama-3.1, Qwen-3-Next-80B; Yang et al., 2025) to extract the principal action in a uniform "to + verb + complement" format. These representations were embedded using all-MiniLM-L6-v2 and clustered with K-means, producing Core Action categories that form the basis for subsequent moral evaluation and generalization. This step ensures the agent learns associations between actions and human judgments rather than spurious similarities in scenario wording.

## 2.4 Generalization Methods

After clustering scenarios with the Probabilistic Context Learner, we introduced a Generalization module to evaluate predictive capacity and provide interpretability. This module uses an LLM to extract descriptive contextual features for each cluster, capturing properties consistently shared within a context but absent elsewhere. Features are represented as concise binary statements, and each scenario is encoded as a vector indicating the presence or absence of these features.

At the cluster level, aggregate feature profiles summarize the characteristic properties of each context. A similarity score between each scenario and context is computed based on these features, and a softmax over scores produces a probability distribution over clusters. The module is trained by minimizing the negative log-likelihood of the true cluster assignments, with feature importance weights learned during training. This allows the model to predict into which cluster a new scenario would be assigned, while simultaneously providing interpretable weights linking individual features to predictive performance. By selecting the most probable label from the assigned cluster distribution, the model predicts the human moral judgment of a scenario, which allows us to evaluate how well COMETH aligns with human judgments. Detailed technical specifications, including model optimization, cross-validation, and exact feature extraction procedures, are provided in the Appendix (see Section A.6).

## 3 Results

We first present the results obtained by the COMETH pipeline using 3 different open-source Large Language Models, namely Mistral-7B-Instruct-v0.3, Llama-3.1-8B-Instruct and Qwen3-Next-80B-A3B-Instruct (Yang et al., 2025). We compare these results to those obtained with end-to-end LLM methods when it is possible to do so.

### 3.1 Pre-processing results

We evaluated the pre-processing algorithm by measuring how well the clustered representations matched the expected core action groups. To assess robustness and reproducibility, we compared the cluster assignments generated by 3 different open-source LLMs across five prompting strategies. Table 1 reports the Adjusted Rand Index (ARI) between the clustering obtained with each LLM and the ideal clustering (50 scenarios per cluster, aligned with the core actions).

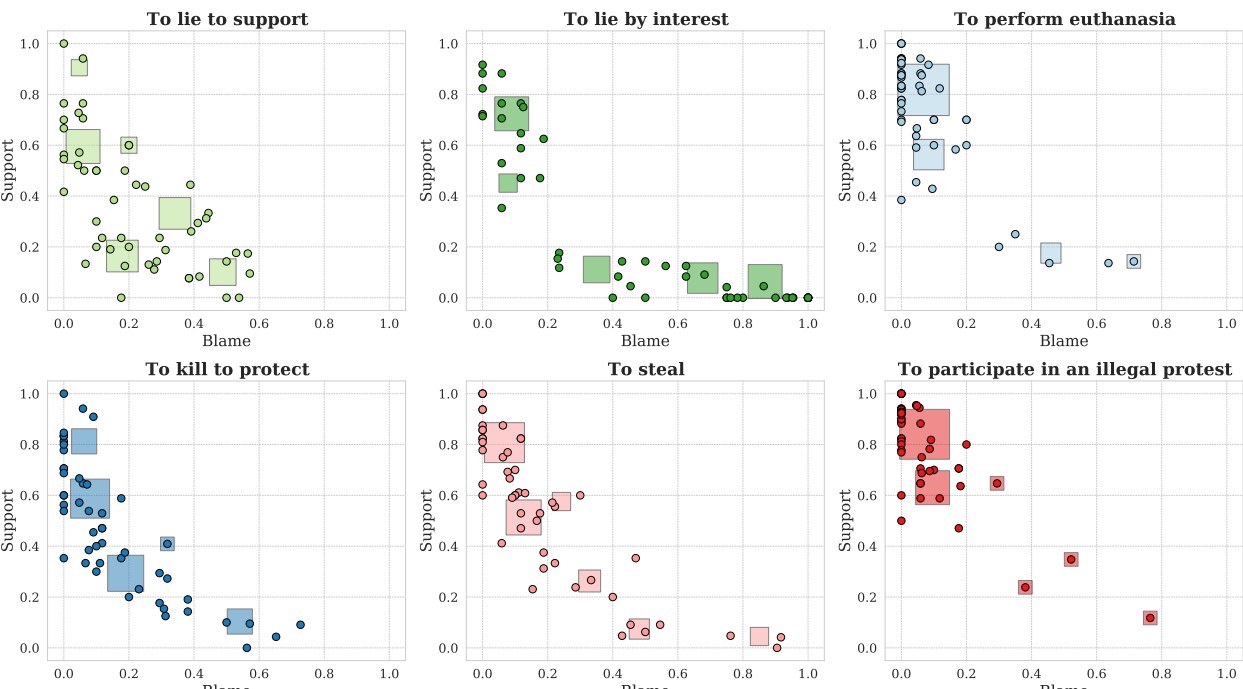

Figure 2: **Clustering of moral scenarios by the Probabilistic Context Learner.** Each subplot corresponds to a specific action type and displays the clustering performed by the agent. Points represent individual states derived from moral scenarios, while squares denote the clusters (or contexts) created by the agent. The size of each square is proportional to the number of states it contains. The coordinates of all elements are based on the probability distribution over the outcomes Support and Blame; the third outcome (Neutral) is implicitly defined as $1 - P(Support) - P(Blame)$, allowing for a two-dimensional representation. The number and shape of the clusters emerge dynamically from the data and align with human intuition based on visual inspection of the same distributions.

For clarity, we show two results: the ARI for the prompt that yielded the highest mean performance across LLMs (referred to as the Main Act prompt), and the best ARI achieved by each LLM individually (prompts are available in the appendix, see A.5.1).

| LLM | ARI Main Act | Best ARI |
|---|---|---|
| Mistral 7B | 0.75 | 0.89 |
| Llama 8B | 0.79 | 0.79 |
| Qwen 80B | 0.87 | 0.90 |

Table 1: Pre-processing accuracy of different LLMs across moral actions (%).

The results in Table 1 show that all three LLMs achieve strong clustering performance, with ARI values up to 0.90. Qwen-80B consistently performs best across prompts, displaying robustness and prompt resilience, while Mistral-7B reaches nearly the same accuracy (0.89) but with higher variability. Llama-8B, by contrast, remains stable but never exceeds 0.79, suggesting limited capacity for fine-grained clustering. Importantly, the choice of prompt substantially affects performance: compact formulations such as Infinitive and OneWord degrade results across all models, whereas MainAct yields consistently high ARIs (0.79–0.87) and thus provides the most reliable and transferable representation. For clarity, Table 1 reports only the best ARI per model and the ARI obtained with the MainAct prompt. The full results, including ARI, NMI, and V-measure across all prompts, are available in Appendix B.1. These findings indicate that the pre-processing step can be implemented reproducibly with different LLMs, but that prompt design is crucial for ensuring semantically coherent action clusters.

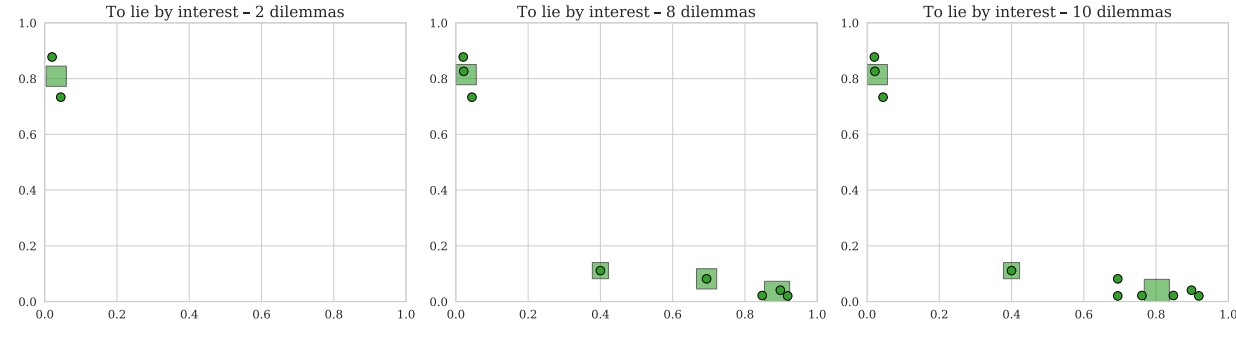

Figure 3: **Temporal evolution of clusters for the action "To lie by interest."** This figure illustrates the agent's continual learning process as new scenarios are introduced. Subplots show successive clustering states over time. When a new scenario is similar to an existing cluster, it is integrated and the cluster is updated (e.g., left panel). If the scenario is dissimilar, a new cluster is created (e.g., center panel, second cluster). When two clusters become sufficiently close, they are merged by the agent (e.g., right panel, last cluster). Clusters containing more scenarios exhibit increased stability and undergo smaller changes during updates.

### 3.2 CLUSTERING RESULTS

We then evaluate the performance of the Probabilistic Context Learner on the set of moral scenarios we presented to the human participants. Figure 2 shows the clustering performed by the agent across the entire dataset. Thanks to pre-processing, each moral scenario is converted into a $(state, action)$ pair, along with a reward distribution derived from our moral judgment survey. For each state, the agent computes a reward distribution and uses it to create *contexts*, which are internal representations grouping similar states, as previously described.

Figure 2 presents one subplot per action to enhance clarity. Each point represents an individual state, while squares denote the clusters (or moral contexts) inferred by the agent. The size of each square reflects the number of states it contains. The coordinates of the elements are based on their distribution values for Support and Blame. Since the third possible outcome, Neutral, is defined as $1 - P(Support) - P(Blame)$, the full distribution can be effectively projected in two dimensions. As illustrated, the number of clusters is not fixed in advance but emerges from the spatial distribution of the states.

As shown in Figure 2, the agent generates a number of clusters that depend on the spatial distribution of the states. This number is not predefined and aligns with the intuitive groupings that a human observer would make when viewing the same distributions as bar plots.

Moreover, the agent exhibits dynamic learning by continuously updating and reorganizing its clusters as new scenarios are introduced. Figure 3 illustrates this process for the action "To lie by interest." on a reduced set of data. When a new scenario closely matches an existing cluster, it is incorporated and the cluster is adjusted accordingly (e.g., Fig. 3-left). If the scenario is dissimilar to all clusters, a new cluster is formed (e.g., Fig. 3-center). When two clusters converge, the agent merges them (e.g., Fig. 3-right). Notably, clusters with more scenarios exhibit greater stability, with their positions less influenced by new data.

### 3.3 GENERALIZATION RESULTS AND COMPARISON TO END-TO-END LLM METHODS

#### 3.3.1 ALIGNMENT RATE COMPARISON

To evaluate the performance of our Generalization module, we report here the alignment rate, defined as the proportion of scenarios for which the model's most probable response matched the majority human judgment. This metric offers an intuitive measure of how closely models reproduce human moral evaluations. Additional evaluation metrics (aligned accuracy and error probability for every action and every prompt) are provided in the Supplementary Materials, see Section B.2 and Section B.3.1.

As shown in Figure 4, end-to-end LLM approaches achieve an average alignment rate of only ∼30%, indicating limited capacity to generalize human moral judgments. In contrast, applying the COMETH pipeline with the same LLMs doubles performance, reaching an average alignment rate of ∼60%. This substantial gain highlights the benefit of grounding predictions in structured feature representations rather than relying solely on direct model outputs. The

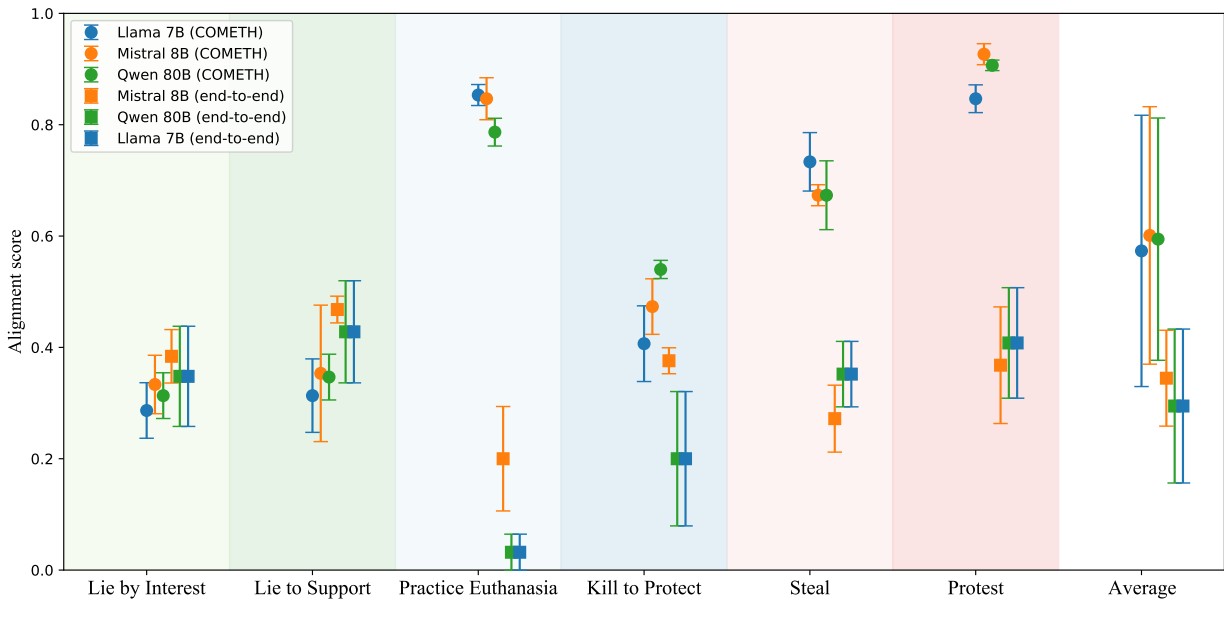

Figure 4: **Comparison of alignment rates between the COMETH pipeline and end-to-end LLMs per action.** Mean alignment rates across the prompts used (3 for the COMETH pipeline, 5 for end-to-end methods) are shown with standard deviations for each action, as well as the overall average across all actions. Results are presented for MistralAI/Mistral-7B-Instruct-v0.3, Meta-LLaMA/LLaMA-3.1-8B-Instruct, and Qwen/Qwen-3-Next-80B-A3B-Instruct (Yang et al., 2025), alongside human baseline data collected via an online survey.

relatively large variance across runs mainly stems from variability in feature extraction quality: when features are less informative, the predictive component of the generalization module deteriorates. Moreover, the diversity of clusters generated by the Probabilistic Context Learner also contributes to fluctuations in efficiency.

Detailed results (Tables S.4 and S.2, Appendix B.3.1) further reveal that performance strongly depends on the type of moral action: unambiguous cases such as *Euthanasia* or *Kill to Protect* show consistently higher alignment than contested actions like *Protest* or *Steal*. They also highlight that while end-to-end models are highly prompt-sensitive (e.g., Llama-8B ranges from 0.41 to 0.64), COMETH reduces this variability and produces more stable clusters across models. Notably, the relative improvement of smaller models such as Mistral-8B suggests that semantic structuring can mitigate scale disparities, making robust moral prediction feasible even with lighter LLMs.

### 3.3.2 INTERPRETABILITY

To complement its clustering performance, the COMETH pipeline also enhances interpretability by revealing how individual features contribute to human moral judgments. Specifically, it assigns weights to features within each cluster, clarifying how the presence of a given element in a scenario shifts its likelihood of being associated with "Support" or "Blame." Figure 5 illustrates this for the action Practice Euthanasia: scenarios mentioning an "approved directive" tend to be assigned to the second cluster, which corresponds to a "Support" judgment. Additional examples are provided in the Appendix (Section B.3.2).

This interpretability constitutes a central contribution of COMETH: beyond improving alignment between LLM predictions and human moral judgments, it offers an explainability framework directly grounded in human data.

## 4 DISCUSSION

This work presents COMETH (Contextual Organization of Moral Evaluation from Textual Human inputs), a framework for context-sensitive moral alignment that integrates a probabilistic context learner, an LLM-based preprocessing stage, and an interpretable generalization module. Compared to end-to-end LLM approaches, COMETH substantially improves performance across multiple LLMs, it approximately doubles the alignment rate with human judgments. The pre-processing step proves highly effective, reliably producing semantically coherent action clusters across diverse

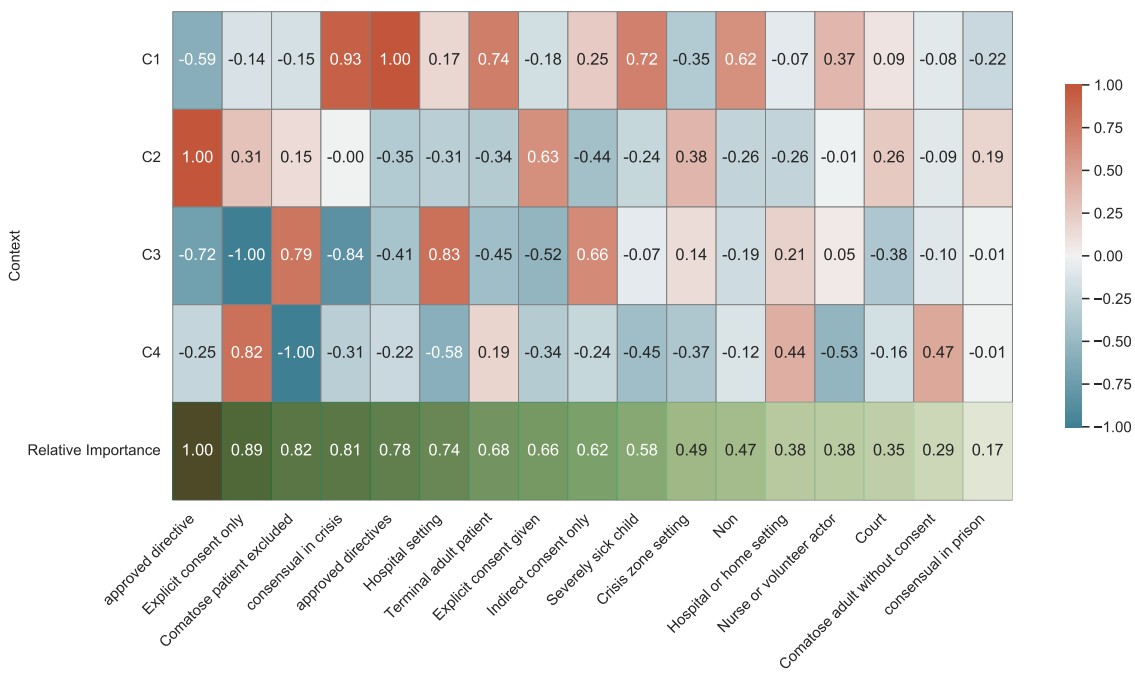

Figure 5: **Feature weights obtained for the action "Practice Euthanasia" with Qwen/Qwen-3-Next-80B-A3B-Instruct (Yang et al., 2025).** The plot reports the relative importance of each feature in shaping cluster assignments, highlighting how different attributes influence whether the scenario is more likely to be assigned to each cluster and then judged with support or blame.

prompts and models. Moreover, the feature-based structure of the pipeline provides interpretable explanations for predictions, contrasting sharply with the opacity of black-box LLM outputs. This interpretability is particularly valuable for smaller or less capable LLMs, which benefit from the structured feature representation to achieve human-aligned moral reasoning that would otherwise be unattainable.

While the model shows meaningful progress in generalization and alignment, limitations remain. COMETH relies on probabilistic representations of moral preference distributions, requires access to human survey data for training, and performs optimally when scenarios follow a consistent syntactic structure. Finally, predictions are evaluated using a majority-label decision rule, whereas calibrated uncertainty estimates or abstention mechanisms would better reflect the inherent ambiguity of moral judgments. These constraints restrict its applicability across the full diversity of naturally occurring moral situations.

Future work should aim to broaden the scope and robustness of COMETH. Scaling to a wider range of scenarios and richer human evaluation datasets could improve generalization. Additionally, integrating mechanisms to automatically select and weight the most informative features would allow the system to achieve maximal alignment across contexts. More ambitious directions include adding active learning to query humans in regions of high uncertainty or disagreement and coupling the pipeline to decision-making modules so that agents can select morally aligned actions while exposing the feature-level rationale.

By doubling the performance of end-to-end LLMs, providing reproducible pre-processing, and enabling interpretable predictions even for smaller models, COMETH offers a concrete and practical path toward building AI systems that can better distinguish and model moral contexts in a structured, transparent, and human-aligned manner. Ultimately, while this work represents weak alignment (Khamassi et al., 2024), it establishes a foundation for stronger alignment strategies that combine human-grounded data, interpretable feature representations, and model-based generalization.

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
