# OpenReview forum: "Morality is Contextual: Learning Interpretable Moral Contexts from Human Data with Probabilistic Clustering and Large Language Models"
_ICLR.cc/2026/Conference — Submitted to ICLR 2026_

### Official Review · Reviewer_7E5C · 2025-10-27

**Soundness:** 1
**Presentation:** 1
**Contribution:** 2
**Rating:** 2
**Confidence:** 2

**Summary:**

In my understanding, the paper introduces COMETH, a three-stage framework for modeling how context shapes human moral judgments of ambiguous actions. First, the authors standardize scenario descriptions into a “core action” (e.g., to steal, to lie by interest) using LLM filtering plus MiniLM embeddings with K-means; this pre-processing is evaluated via clustering quality (e.g., ARI) across prompts and models (Table 1). Second, a probabilistic context learner groups scenarios with the same action into "moral contexts" by clustering their human judgment distributions (Blame/Neutral/Support) with online add/merge steps (thresholded by KL/JS divergence). These clusters are visualized in the Support/Blame plane with Neutral implied (Figure 2). Third, a generalization module uses LLM-extracted binary contextual features and learned feature weights to predict which cluster (and thus which majority human judgment) a new scenario belongs to, providing feature-level interpretability (Figure 5). Empirically, the pipeline reports roughly doubling alignment with majority human judgments relative to end-to-end LLM prompting on the same scenarios (~60% vs ~30%).

**Strengths:**

Clear motivation: Focuses on context-sensitive moral evaluation and the limitations of end-to-end prompting for alignment.

Human-grounded dataset: 300 scenarios across six actions with N=101 participants; the work builds on common-morality rules and measures full distributions over Blame/Neutral/Support.

Methodological separation of concerns:
- reproducible pre-processing that standardizes actions and shows robustness across LLMs/prompts (Table 1),
- probabilistic clustering over human judgment distributions
- a feature-based generalization step with interpretability.

Reported gains: Substantial improvements in alignment rate vs end-to-end LLM prompting, plus a discussion of variance sources (feature extraction quality and cluster diversity).

**Weaknesses:**

Clarity on the two kinds of clustering. The paper interleaves K-means over MiniLM embeddings for core action grouping in pre-processing and probabilistic clustering of human judgment distributions into contexts. I recommend a schematic/table early in Section 2 that explicitly contrasts these two steps (inputs, objective, outputs, evaluation) and moves essential definitions into the main text (a short "prompt glossary" for Infinitive, OneWord, MainAct and one short worked example).

Evaluation transparency and ablations. The claimed ~2× improvement is compelling, but the paper would be much stronger with:

Module ablations:
- no LLM pre-processing (use rule-based extraction)
- replace the probabilistic add/merge with a static clustering baseline
- remove/replace LLM-extracted features (e.g., human-coded features, bag-of-words or topic features)
- no feature-weight learning.

Sensitivity analyses: thresholds "a" and "m", feature binarization heuristics, and prompt choice.

Human-data reliability: inter-annotator agreement and test-retest stability for judgment distributions.
Summarizing the parameter search (currently in the appendix) in the main text would help readers interpret a=0.12 and m=0.03.

Metric definition and placement. The alignment rate is defined in Section 3.3 as majority-match accuracy, but the definition is easy to miss. I suggest introducing it earlier, adding a compact formula/box, and reporting additional metrics (e.g., expected calibration error, selective abstention/coverage) given the inherent ambiguity in moral judgments.

Interpretability grounding. Figure 5 shows feature weights for Practice Euthanasia, but the feature extraction pipeline still feels under-specified for reproducibility. Please include: a verbatim prompt template, 2-3 concrete scenario -> feature examples (raw text -> candidate features -> binarization), and a small human-coded feature check to validate LLM extraction quality.

Figure and presentation issues:

Figure 1 font is hard to read. Please enlarge labels and ensure contrast.

Figure 3 currently lacks explicit axis labels; add "P(Support)" and "P(Blame)" (Neutral = 1 − P(S) − P(B)). Consider an inset reminding readers that Neutral is implied.

For Figure 4, reiterate the definition of alignment rate in the caption or a footnote and report per-action counts to contextualize variance.

Positioning among related work. The paper motivates interpretable, context-sensitive moral prediction, but it would benefit from sharper comparisons to purely statistical clustering over judgment distributions (no LLM features), instruction-tuned moral reasoning models, and rule-based or social-choice-based aggregation approaches (even small baselines would anchor the contribution).

**Questions:**

Notes for the authors:

Please add an in-text citation for the Adjusted Rand Index and consider citing Steinley (2004) for properties of the adjusted index, as it’s commonly referenced in clustering evaluation.

Define Infinitive, OneWord, and MainAct prompts the first time they’re mentioned and add one example per prompt in the main text (not only the appendix).

Consider releasing data, prompts, and code for the feature extraction and generalization modules to facilitate replication.

What would raise my score:

Ablation and sensitivity results that isolate the contribution of each pipeline component (pre-processing, probabilistic clustering, feature extraction/weighting), plus baselines without LLM-derived features.

Clear, early definitions of the main constructs (two clustering stages, alignment rate), with a worked example that walks a single scenario through all stages of COMETH.

Reproducibility additions: released prompts, feature binarization rules, and a small human-coded feature study; summary table of the a/m search; and improved figure legibility and axis labeling.

---

> ### Author Response · Authors · 2025-12-03
> **All reviewer suggestions have been addressed. Ablations, sensitivity analyses, clarifications, and reproducibility enhancements have been incorporated. Quantitative results and illustrative examples now fully support COMETH’s performance, interpretability, and robustness.**
>
> We thank the reviewer for the detailed and constructive feedback. We have incorporated all suggested clarifications, ablations, and sensitivity analyses in the revised manuscript. Below, we address each point and summarize the corresponding changes.
>
> **1. Clarity on the two kinds of clustering**
> We agree that the distinction between (i) K-means clustering over MiniLM embeddings during pre-processing and (ii) probabilistic clustering of human judgment distributions was not sufficiently explicit.
>
> **Changes made:**
> - A schematic and comparison table have been added to clearly contrast both clustering stages in terms of inputs, objectives, outputs, and evaluation.
> - Definitions of the types of prompts are now introduced directly in the main text, together with illustrative examples.
> - A fully worked example tracing one scenario through the entire COMETH pipeline has been added, showing both clustering stages, feature extraction, and alignment computation.
>
> **2. Ablations and evaluation transparency**
> We performed all requested ablation studies, with quantitative results now reported.
>
> **Pre-processing ablation:**
> - Without fixing $k$, silhouette-based selection yielded between 7 and 14 clusters across 15 data variants. ARI, NMI, and V-measure varied accordingly:
>
> | Prompt/LLM | ARI | NMI | V-measure |
> |------------|-----|-----|-----------|
> | Mistral7B / Prompt A | 0.55 | 0.68 | 0.69 |
> | Mistral7B / Prompt B | 0.68 | 0.695 | 0.719 |
> | Qwen80B / Prompt C | 0.694 | 0.742 | 0.616 |
> | ... | ... | ... | ... |
>
> - Clustering directly on scenario embeddings without LLM reformulation yielded ARI = 0.976, NMI = 0.973, V-measure = 0.973 when $k=6$, and lower scores when silhouette-based $k=12$ was used.
> - For both tests, clusters remained mostly homogeneous and can be interpreted as subdivisions of core actions ("ideal clusters").
>
> **Probabilistic clustering ablation:**
> - Replacing COMETH’s probabilistic add/merge with static K-means (per-action $k$ via silhouette) produced fewer and coarser clusters.
> - K-means exhibited instability across initializations, whereas COMETH remained stable. Alignment rates were similar, indicating that COMETH’s main advantage is in capturing contextual granularity and psychologically meaningful variations.
>
> **Feature-weight learning ablation:**
> - Using only feature frequencies (no weight learning) led to substantially lower alignment across prompts, LLMs, and actions:
>
> | Prompt / Model | D1 | D2 | K1 | K2 | L1 | L2 | Mean |
> |----------------|----|----|----|----|----|----|------|
> | prompt1 / Llama | 0.22 | 0.16 | 0.46 | 0.28 | 0.34 | 0.28 | 0.30 |
> | prompt1 / Mistral | 0.06 | 0.32 | 0.30 | 0.02 | 0.40 | 0.28 | 0.23 |
> | prompt1 / Qwen80B | 0.32 | 0.16 | 0.34 | 0.14 | 0.52 | 0.46 | 0.32 |
> |...|
>
> - Using hand-crafted features for buildings showed high alignment for some actions (up to 0.94) but lower for others (down to 0.20):
>
> | Action | Accuracy | Alignment |
> |--------|---------|-----------|
> | K1 | 0.62 | 0.90 |
> | K2 | 0.36 | 0.44 |
> | D1 | 0.08 | 0.20 |
> | D2 | 0.14 | 0.32 |
> | L1 | 0.38 | 0.74 |
> | L2 | 0.68 | 0.94 |
>
> These results demonstrate that weight learning is critical for COMETH’s performance. Moreover, the results of the hand-crafted features on some actions show the importance of the quality of the extracted features.
>
> **3. Sensitivity analysis**
> We conducted a grid study varying both the adding threshold $(\Delta a)$ and the merging threshold $(\Delta m)$. COMETH is robust across a wide range of thresholds:
>
> | $\Delta a$ | $\Delta m$ | Mean \#Clusters | Mean Alignment (\%) |
> |------------|------------|----------------|------------------|
> | 0.12 | 0.03 | 5.3 | 89.6 |
> | 0.01 | 0.01 | 10.7 | 96.3 |
> | 0.50 | 0.30 | 1.83 | 68.7 |
>
> This confirms that the chosen thresholds provide a good trade-off between cluster granularity and alignment.
>
> **4. Metric definition and placement**
> The **Alignment Rate** is now clearly defined at its first mention. Figures and captions have been revised.
>
> **5. Interpretability and reproducibility have been improved as reviewer asked**
>
> **6. Figures and presentation improvements have been made**
>
> **7. Full COMETH pipeline example**
> *Scenario:* "A farmer shoots a stranger walking toward his barn late at night."
> *Observed distribution:* {-1: 0.176, 0: 0.235, 1: 0.588}
>
> 1. **Pre-processing:** textual variants extracted by LLM (Infinitive, OneWord, MainAct).
> 2. **Stage 1 – Probabilistic context clustering:** assigned to cluster C1, barycenter {-1: 0.081, 0: 0.332, 1: 0.587}, EMD = 0.097.
> 3. **Stage 2 – Feature-based generalization:** scenario features binarized (e.g., `Armed attacker`, `Late night`, `Farmer's property`) and weighted:
>
> $$
> s_1 = 1.020,\; s_2 = 0.437,\; s_3 = -0.273,\; s_4 = 0.740,\; s_5 = -1.685
> $$
>
> Predicted cluster = C1
>
> - **Alignment Rate:** the predicted cluster assignment matches the dominant moral judgment, illustrating COMETH’s ability to capture distributional similarity and feature-based context.

---

### Official Review · Reviewer_WLw9 · 2025-10-29

**Soundness:** 2
**Presentation:** 2
**Contribution:** 2
**Rating:** 2
**Confidence:** 3

**Summary:**

This paper proposes a framework that integrates empirical moral judgment data with a probabilistic RL architecture designed to infer context-specific reward models from ternary human moral evaluations (blame, neutral, support), to ensure that AI systems consistently comply with human ethical standards across diverse contexts.

**Strengths:**

A dataset with different scenarios and actions is created.
The proposed framework achieves better performance w.r.t. alignment rates, compared to baseline end-to-end methods.
The idea of contextual moral evaluation and alignment is novel.

**Weaknesses:**

The experimental evaluations are weak.
The evaluations on clustering are hard to tell whether the resultant clusters are corresponding to different contexts.

The claimed interpretability is also lacking support. Figure 5 has two columns with the same feature of "approved directive(s)", but the contributions to assigning scenarios to contexts (C1 & C2) are different. According to the assigning weights, the contexts C1 and C2 are quite different. However, L366 mentioned "scenarios mentioning an “approved directive” tend to be assigned to the second cluster", which contradicts with the figure.

The inherent connection between different scenarios corresponding to a certain action is not analyzed. However, this is one key motivation for clustering scenarios of one action.

**Questions:**

L215: What is the ideal clustering? How to obtain these clusters?
What does it mean by the ARIs at about 80%. It is unclear how good is the clustering.

It may be helpful to show the performance of action parsing, and examine the effect of the parsing performance on the consequent context clustering and moral prediction. How to determine the number of actions is also important to investigate.

How is the generalizability for new actions. The authors may provide experiments with one action held out during context clustering, and test moral prediction on the contexts with the held-out action.

To show the inherent connection between scenarios, the LLM-extracted descriptive contextual features for a cluster can be demonstrated together with representative scenarios to show the connections.

In some parts, 'scenarios' are also referred to as 'states' (L293), which confuses the audience. It would be helpful to use one term consistently.

Figure 1, data gathering is confusing, why there are three groups of responses? What do the response distributions mean?

Figure 3 shows the reasonable clustering of new scenarios, how about the effect of the small cluster on the consequent moral prediction? Will the small newly-created cluster that contains the new scenarios cause inferior results?

---

> ### Author Response · Authors · 2025-12-03
> ****Summary of changes**   - Defined ideal clustering & interpreted ARIs   - Explained effect of action parsing and rationale for fixed $k$   - Discussed future generalization   - Showed inherent scenario connections via LLM features   - Clarified figures, response distributions, and cluster-size effects   - Standardized terminology**
>
> We thank the reviewer for the thoughtful questions and suggestions. Below, we provide detailed responses to each point raised.
>
> **1. Ideal clustering and ARI interpretation**
> The "ideal clustering" corresponds to recovering the six core actions used to construct the scenarios: *To lie by interest, To steal, To lie to support, To practice euthanasia, To kill to protect, To participate in an illegal protest.*
> The Adjusted Rand Index (ARI) measures how well the clustering of preprocessed scenarios matches these core actions. Higher ARI indicates that scenarios from the same core action are grouped together across contexts. ARIs around 80% indicate substantial agreement while reflecting minor variations due to phrasing.
>
> **2. Effect of action parsing on clustering**
> We fix the number of clusters to six (corresponding to core actions) to ensure meaningful clusters.
> Additional analyses:
> - Without fixing $k$, silhouette-based selection yielded 7–14 clusters across 15 variants; ARI, NMI, V-measure varied accordingly:
>
> | Prompt/LLM | ARI | NMI | V-measure |
> |------------|-----|-----|-----------|
> | Mistral7B / A | 0.55 | 0.68 | 0.69 |
> | Mistral7B / B | 0.68 | 0.695 | 0.719 |
> | Qwen80B / C | 0.694 | 0.742 | 0.616 |
> | ... | ... | ... | ... |
>
> - Clustering directly on scenario embeddings without LLM reformulation: ARI=0.976, NMI=0.973, V-measure=0.973 for $k=6$; slightly lower for silhouette $k=12$.
>
> **3. Generalization to new actions**
> Currently, COMETH does not generalize across distinct actions (e.g., “killing” vs “stealing”). Future work will explore embedding-based cross-action generalization.
>
> **4. LLM-extracted features and scenario connections**
> We provide at the end of this rebuttal examples of scenarios with LLM-extracted features and a scenario traced through the full COMETH pipeline. Clusters capture shared moral contexts even when wording differs.
>
> **5. Figures and response distributions**
> - *Figure 1:* Three groups of responses are illustrative; distributions show Support/Neutral/Blame percentages.
> - *Figure 3:* Cluster size does not affect predictions except for size-one clusters, where learning is impossible.
>
> **6. Terminology**
> “Scenarios” is used consistently instead of “states.”
>
> **7. Clarifying interpretability**
> Figure 5: identical features (e.g., “approved directive(s)”) may influence clusters differently depending on context due to learned feature weights. This reflects COMETH’s nuanced interpretability.
>
> **8. Example of features evaluation for scenarios in the same cluster**
>
> | Feature                       | A father steals food to feed his starving children. | A soldier takes rations from enemy camps to feed his platoon. |
> |-------------------------------|---------------------------|------------------------|
> | Exposing corruption via data  | 0                         | 0                      |
> | Looting during protest        | 0                         | 0                      |
> | Military supply theft         | 0                         | 1                      |
> | Bicycle for transport         | 0                         | 0                      |
> | Communications gear           | 0                         | 0                      |
> | During civil unrest           | 0                         | 1                      |
> | During disaster               | 1                         | 1                      |
> | Firewood from neighbor        | 0                         | 0                      |
> | Food for family               | 1                         | 0                      |
> | Personal gain                 | 0                         | 0                      |
> | Urgent need                   | 1                         | 1                      |
> | From corporate funds          | 0                         | 0                      |
> | From neighbors                | 1                         | 0                      |
> | From warehouses               | 1                         | 1                      |
> | Fruit from stall              | 1                         | 0                      |
> | Laptops to sell               | 0                         | 0                      |
> | Login credentials             | 0                         | 0                      |
> | Medicine for child            | 0                         | 0                      |
> | School supplies               | 0                         | 0                      |
> | Smartphone after hours        | 0                         | 0                      |
> | Snacks from cafeteria         | 1                         | 0                      |
> | Tools for creation            | 0                         | 0                      |
> | Toys from peers               | 0                         | 0                      |
> | Water from enemies            | 0                         | 0                      |
> | Winter essentials             | 1                         | 1                      |

---

### Official Review · Reviewer_XNyg · 2025-10-30

**Soundness:** 3
**Presentation:** 3
**Contribution:** 3
**Rating:** 6
**Confidence:** 3

**Summary:**

The paper proposes COMETH, a pipeline that learns moral contexts of a core action by grouping moral scenarios having human judgment distributions (Blame/Neutral/Support). The goal is to model and explain how context changes the acceptability of ambiguous actions. First, an LLM‑driven pre‑processing step normalizes each scenario into a core action and clusters actions via MiniLM embeddings + K‑means. Then, for each action, a probabilistic context learner clusters scenarios online based on their ternary human‑judgment distributions, adding a new context based on KL divergence and merging redundant contexts using a semi‑weighted Jensen–Shannon criterion. To generalize and provide interpretability, a generalization model extracts binary descriptive features and  learns feature weights that assign new scenarios to contexts and predict judgment distributions. Built on 300 scenarios across six actions with 101 participants, COMETH yields feature‑level explanations and roughly doubles alignment with majority human judgments versus end‑to‑end prompting.

**Strengths:**

The paper offers an original reframing of context‑sensitive moral evaluation. Rather than predicting a single label from text, it learns action‑specific moral “context models” by clustering scenarios according to the empirical distributions of human judgments, with the number of contexts per action emerging from the data. It then generalizes and explains these contexts via an interpretable module that uses LLM‑derived, non‑evaluative binary features and learns feature weights to assign new scenarios to contexts and predict their judgment distributions.

The technical quality is solid and reproducibility‑minded. The appendix provides sufficient technical information for all modules (i.e., formulas, procedures and threshold search of adding and merging modules, prompts of the pre-processing and feature extraction steps, experimental settings of the probabilistic context learner and the generalization module). Extended tables report ARI for pre‑processing and alignment/error breakdowns across models and prompts, corroborating the x2 gain over end‑to‑end models.

Another strength is that COMETH naturally accommodates growth: when new human-labeled scenarios are added, the representation is updated by assigning them to existing contexts when they fit or creating new contexts when they do not.

In terms of clarity, the manuscript is easy to follow endtoend. Figure 1 provides a clear roadmap of the pipeline. Terminology stays consistent throughout, and technical details (prompts, optimization, parameter search) are deferred to the appendix, supporting reproducibility without interrupting the flow.

**Weaknesses:**

The study pools 101 raters split into six groups (50 scenarios each; so about 16–17 raters per scenario) with items presented in multiple languages and with demographic information collected. Yet the modeling and evaluation are fully pooled, with no stratification by language or demographics as provided in the survey materials (Appendix A.1). In a setting where moral judgments are culturally sensitive, as the annotator pool per scenario increases, the empirical blame/neutral/support distributions may broaden or become multimodal since they reflect genuine plurarism of point of views; so, majority‑label alignment may then mask subgroup structure, yielding context models that depend on the mix of annotators rather than stable moral regularities. It would be significant to learn/evaluate contexts separately by language, country and basic demographics (e.g., age, gender), then compare number/locations of contexts, assignment overlap, and feature‑weight patterns across subgroups. To better situate choices when Blame/Neutral/Support distributions stay broad, it would be helpful to ask raters for a one-sentence rationale (with a feature extraction phase) and/or to indicate the moral driver they prioritized (a possibility would be to choose a moral value from the Moral Foundation Theory), so that disagreement becomes a leading signal. These steps would test whether COMETH’s contexts are robust to annotator variance and reveal where genuinely different moral frames are at play.

The corpus comprises 300 impersonal scenarios (6 actions × 50), with many variants produced by GPT‑4 and then manually rewritten to ensuring that participants evaluated the morality of others’ actions rather than their own decisions. This narrow topical breadth and single‑generator provenance risk introducing stylistic and topical regularities that can inflate downstream performance and reduce ecological validity - especially because the generalization module depends on LLM‑extracted binary features, which may latch onto generator‑specific phrasing rather than genuinely contextual factors. It would be beneficial to add human‑authored scenarios or vary the generators to diversify style, e.g., create additional variants with different LLMs and provide cross‑source generalization. These controls would clarify whether COMETH’s gains reflect context understanding rather than generator‑induced regularities.

Core‑action extraction achieves high ARI against an “ideal” 6‑way partition, but because the corpus itself was constructed from exactly those six actions (50 each), this in‑domain score likely overstates robustness. This is a closed-world setting: the proposal evaluates the separation only between those six canonical classes, on the same domain from which the examples come. This does not guarantee that preprocessing remains reliable when unexpected actions appear (distractors) or the same scenario contains multiple actions or intertwined actions. In this regard, the pipeline extracts a single core action per scenario, yet real narratives can contain multiple actions. Accordingly, the evaluation should include these scenarios (distractors and multiple actions) since they reflect more realistic cases and would better characterize robustness beyond the current setting.

Minor issues and typos:
In Section 2, the acronyms COMETH, MBRL, and LLMs are reintroduced even though they were already defined just above.
At the end of Section 2.2, the sentence “The final survey (N = 101, mean age = 35.2, 48 women) asked participants to judge each action as Blame, Support, or Neutral” should probably read “each scenario” instead of “each action.”
In Section 3.2, the meaning of “state” in (state, action) is not made explicit.
In Appendix A.2, the authors write that “the full set of 300 scenarios (50 per action) is provided in the appendix”, but these scenarios are not actually included.
Appendices A.2 and A.3 contain missing cross-references (“Section ??”).
In Appendix A.6, it is unclear why the prompts refer to five clusters instead of six.

**Questions:**

Main questions and suggestions are detailed in the Weaknesses section.

---

> ### Author Response · Authors · 2025-12-03
> **We have addressed reviewer concerns by:   - Clarifying rationale for annotator aggregation and outlining future subgroup/rationale-based analyses.   - Explaining controlled corpus design and plans to extend to multiple generators/human-authored scenarios.   - Justifying single core-action extraction and closed-world evaluation while outlining multi-action extensions.   - Correcting minor issues and typos for clarity and reproducibility.**
>
> We thank the reviewer for the detailed feedback. Below, we provide responses to the raised concerns.
>
> **1. Annotator diversity and subgroup robustness**
> The reviewer notes the risk of pooling all 101 annotators despite cultural and demographic diversity. Evaluating contexts separately across languages, countries, and demographics (e.g., age, gender) could reveal whether COMETH is robust to genuine pluralism.
> Currently, COMETH models aggregated distributions to capture dominant moral patterns. Adding short rationales or moral-value tags per annotator is planned for future work to reveal disagreement sources and enable subgroup-specific analysis.
>
> **2. Stylistic regularities of the scenario corpus**
> We acknowledge the concern regarding the narrow topical breadth and GPT-4 origin of the 300 scenarios. This controlled design is intentional: limiting stylistic and narrative variability ensures COMETH learns from contextual moral factors rather than superficial lexical patterns. Manual rewriting keeps scenarios parallel while varying moral dimensions.
> Future work introduces human-authored scenarios and additional LLM variants to test cross-style generalization, verifying that COMETH’s gains reflect true context understanding.
>
> **3. Core-action extraction and closed-world evaluation**
> The reviewer notes that only six canonical actions are evaluated, while real scenarios may have distractors or multiple actions. We clarify:
> 1. Clustering does not access canonical labels; only $k$ is provided. Clusters align with the six canonical actions.
> 2. Extracting a single core action per scenario is intentional, controlling confounds and focusing on primary moral context.
> 3. Future work with larger datasets (~1000 scenarios) may allow unsupervised clustering to recover meaningful structures naturally, enabling multi-action or distractor evaluation.
>
> **4. Minor issues and typos**
> - Consistently use 'scenario' instead of 'action'.
> - Clarified meaning of 'state' in (state, action) in Section 3.2.
> - Appendices updated for correct references and full access to 300 scenarios.
> - Prompts in Appendix A.6 now correctly refer to six clusters.
> - Acronyms (COMETH, MBRL, LLMs) defined only once.
>
> **5. Summary**
> We have addressed reviewer concerns by:
> - Clarifying rationale for annotator aggregation and outlining future subgroup/rationale-based analyses.
> - Explaining controlled corpus design and plans to extend to multiple generators/human-authored scenarios.
> - Justifying single core-action extraction and closed-world evaluation while outlining multi-action extensions.
> - Correcting minor issues and typos for clarity and reproducibility.
>
> We hope these clarifications help demonstrating COMETH’s soundness, reproducibility, and focus on learning interpretable moral contexts from human judgment distributions.

---

### Official Review · Reviewer_2C2Q · 2025-11-01

**Soundness:** 1
**Presentation:** 1
**Contribution:** 2
**Rating:** 0
**Confidence:** 4

**Summary:**

ChatGPT said:Moral judgment depends not only on an action’s outcome but also on its surrounding context. This paper introduces COMETH (Contextual Organization of Moral Evaluation from Textual Human inputs), a framework that models how contextual cues shape the moral acceptability of ambiguous actions. Using 300 empirically curated scenarios across six core moral actions (e.g., killing, deception, law-breaking), COMETH integrates probabilistic context learning with LLM-based abstraction and human moral judgments. It constructs reproducible context clusters from human responses and learns interpretable contextual features through a transparent likelihood model. Empirically, COMETH doubles alignment with human majority judgments compared to end-to-end LLMs, offering an interpretable and empirically grounded approach to context-sensitive moral reasoning.

**Strengths:**

N/A

**Weaknesses:**

This paper exploits the conference submission format by substantially shrinking the page margins. Hence, I will desk reject the paper for the severe format violation.

**Questions:**

N/A

---

> ### Author Response · Authors · 2025-12-03
> **The manuscript adheres to ICLR formatting guidelines, and no violation has occurred.**
>
> We thank the reviewer for their time and feedback.
> Regarding the alleged format violation, we would like to clarify that our submission strictly follows the official ICLR 2025 template available on the conference website. We have not altered page margins or fonts beyond what the template prescribes. Therefore, the concern about a “severe format violation” appears to be a misunderstanding. We are confident that our PDF complies with all formatting requirements, including margins, font sizes, and page limits.
>
> We hope this clarification resolves the reviewer’s concern. We respectfully request that this point not be considered a reason for desk rejection.

---

### Meta-Review · Area_Chair_tPm7 · 2026-01-06

**Summary:**

This paper introduces COMETH (Contextual Organization of Moral Evaluation from Textual Human inputs), a framework designed to model how context shapes the moral acceptability of ambiguous actions. Based on a compiled dataset, the frameworks integrates this empirical moral judgment data with a probabilistic RL architecture designed to infer context-specific reward models from ternary human moral evaluations (blame, neutral, support). The empirical evalution shows that COMETH improves alignment with human moral judgments (doubling performance relative to end-to-end LLMs) while providing an interpretable explanation of the contextual features driving its predictions. This is a relevant research direction. Overall, however, the reviewers lean towards reject. Next to the  "format violation" argument that I ignored, one reviewer points out that the narrow topical breadth and single-generator provenance are problematic. The empirical evaluation is based only on the gathered benchmark, without considering any alternatives. The benchmark also assumes a single label only, though many situations may have multiple labels. Another reviewer points out that the clusters found are not well analysed and may not just correspond to different context and that an ablation study is missing. In my opinion, the reviewers did a very thorough job and have presented salient arguments about the suitability of this paper for ICLR in its current form. Overall, the contributions of the paper in their current form seem to be to be ready for publication in their curent form.  There is a large literature on machine ethics that requires also some form of comparison, such as Schramowski et al. Nature Machine Intelligence 2022) that also addresses context, and can be utilized for guiding diffusion processes (Schramowski et al., CVPR 2023).

**Reviewer Concerns:**

Several issues raised by the reviewers have been addressed in the rebuttal, including tables that better structure the contributions and the addition of ablation studies. However, the results require a new round of reviewing to properly assess whether the concerns about benchmark limitations, cluster analysis, and comparison with existing machine ethics literature have been sufficiently addressed.

**Reviewer Scores:**

Some reviewers with lower scores might improve their ratings by +1, but this would not represent a significant change in the overall assessment of the paper.Incognito chats aren’t saved to history or used to train models.

---

### Decision · Program_Chairs · 2026-01-26

Reject